# Long-Term Consequences of Asymptomatic SARS-CoV-2 Infection: A Systematic Review and Meta-Analysis

**DOI:** 10.3390/ijerph20021613

**Published:** 2023-01-16

**Authors:** Yirui Ma, Jie Deng, Qiao Liu, Min Du, Min Liu, Jue Liu

**Affiliations:** 1Department of Epidemiology and Biostatistics, School of Public Health, Peking University, Beijing 100191, China; 2Institute for Global Health and Development, Peking University, Beijing 100871, China

**Keywords:** COVID-19, SARS-CoV-2, asymptomatic, symptomatic, long-term consequence, systematic review, meta-analysis

## Abstract

Little is known about the long-term consequences of asymptomatic infection caused by severe acute respiratory syndrome coronavirus 2 (SARS-CoV-2). We aimed to review the data available to explore the long-term consequences of asymptomatic SARS-CoV-2 infection in the real world. We searched observational cohort studies that described the long-term health effects of asymptomatic SARS-CoV-2 infections. Random-effects inverse-variance models were used to evaluate the pooled prevalence (PP) and its 95% confidence interval (CI) of long-term symptoms. Random effects were used to estimate the pooled odds ratios (OR) and its 95%CI of different long-term symptoms between symptomatic and asymptomatic infections. Five studies involving a total of 1643 cases, including 597 cases of asymptomatic and 1043 cases of symptomatic SARS-CoV-2 infection were included in this meta-analysis. The PPs of long-term consequences after asymptomatic SARS-CoV-2 infections were 17.13% (95%CI, 7.55–26.71%) for at least one symptom, 15.09% (95%CI, 5.46–24.73%) for loss of taste, 14.14% (95%CI, −1.32–29.61%) for loss of smell, and 9.33% (95%CI, 3.07–15.60) for fatigue. Compared with symptomatic SARS-CoV-2 infection, asymptomatic infection was associated with a significantly lower risk of developing COVID-19-related sequelae (*p* < 0.05), with 80% lower risk of developing at least one symptom (OR = 0.20, 95%CI, 0.09–0.45), 81% lower risk of fatigue (OR = 0.19, 95%CI, 0.08–0.49), 90% lower risk of loss of taste/smell (OR = 0.10, 95%CI, 0.02–0.58). Our results suggested that there were long-term effects of asymptomatic SARS-CoV-2 infection, such as loss of taste or smell, fatigue, cough and so on. However, the risk of developing long-term symptoms in asymptomatic SARS-CoV-2 infected persons was significantly lower than those in symptomatic SARS-CoV-2 infection cases.

## 1. Introduction

Since the outbreak of coronavirus disease 2019 (COVID-19), the worldwide pandemic of severe acute respiratory syndrome coronavirus 2 (SARS-CoV-2) has brought enormous challenges and burdens to the global economic and healthcare system [1]. Globally, as of 18 November 2022, more than 633.6 million confirmed cases of COVID-19 have been reported, including more than 6.60 million deaths, according to the World Health Organization (WHO) [2]. In the original classification, the severity of COVID-19 was divided into four categories, including mild, moderate, severe, and critical cases [3]. However, with the persistent pandemic of COVID-19, more and more evidence has shown that quite a number of SARS-CoV-2 infection cases were asymptomatic but they were infectious as symptomatic as infection cases, which is posing a challenge to the prevention and control of COVID-19 epidemic [3]. Asymptomatic SARS-CoV-2 infections were defined as individuals who did not present any symptoms at the time of SARS-CoV-2 testing or diagnosis but tested positive [4]. Jingjing He et al.’s study, which included a total of 50,155 participants [5], suggested that the pooled proportion of asymptomatic SARS-CoV-2 infection was 15.6%. Qiuyue Ma et al. conducted a systematic review and meta-analysis of 95 unique studies involving nearly 29.78 million people, and the results showed that the pooled proportion of asymptomatic infections among the tested people was 0.25%, and among people with identified COVID-19 was 40.50%; the high proportion of asymptomatic SARS-CoV-2 infections emphasizes the potential risk of contagion of asymptomatic infections in communities [4].

Existing studies suggested that while most patients have recovered from COVID-19, for a large amount of persons, whether they were male or female, hospitalized or not, old or young, the virus has caused a range of continuous effects or post-infection sequelae [6]. According to the definition by Delphi consensus, post-COVID-19 condition, also known as long COVID-19, refers to the condition in which an individual with a history of probable or confirmed SARS-CoV-2 infection, usually three months from the onset, has some symptoms that persist for at least two months and cannot be explained by another diagnosis, has become an important target for research and clinical practice [7]. Familiar long COVID-19 symptoms include, but are not limited to, fatigue, shortness of breath, and cognitive dysfunction, and often have some adverse effects on daily functioning [7,8]. An existing systematic review and meta-analysis showed that 63.87% of COVID-19 patients developed at least one COVID-19-related long-term effect at 6–12 months after recovery or discharge, and 58.89% of patients still suffered these impacts at 12 months and above [8]. Previous studies have suggested that the long-term sequelae of COVID-19 were associated with the severity of SARS-CoV-2 infection at the time of onset—the more severe the disease, the higher the risk of sequelae and the more severe the sequelae [9,10,11]. The proportion of long-term sequelae in hospitalized COVID-19 patients was significantly higher than that in non-hospitalized patients, and the COVID-19 sequelae in intensive care unit (ICU) patients were more serious [10]. In addition, it has been suggested that the SARS-CoV-2 vaccination could significantly reduce the risk and severity of COVID-19 sequelae [9,12,13].

Research about the long-term consequences of asymptomatic SARS-CoV-2 infection has aroused wide interest among scholars globally, but the conclusions of different studies have varied. The results of a matched cohort study in Scotland by Claire E. Hastie et al. showed that no COVID-19-related sequelae were observed among asymptomatic SARS-CoV-2 infection patients [9]. However, Maddalena Peghin et al.’s research showed that nearly 5.4% of asymptomatic SARS-CoV-2 infection patients developed at least one post-COVID-19 related symptom in a six-month assessment after acute SARS-CoV-2 infection onset [14]. A study performed by Mei Zhou et al. showed that some asymptomatic COVID-19 patients might develop a certain degree of abnormal computed tomography (CT) or lung function injury, but were less common and prominent than those severe or critical recovered patients (RPs) in the 3 months follow-up after recovery [15]. A review by Anna Malkova et al. about Post-COVID-19 Syndrome (PCS) among asymptomatic/mild SARS-CoV-2 infection patients also suggested that nearly an average of 30–60% of patients, especially females, developed PCS, with fatigue, shortness of breath, cough, and anosmia the most frequent symptoms [16]. Recent studies have shown that children with mild or asymptomatic SARS-CoV-2 infection might incur some long-term sequelae, such as lethargy, fatigue, and cough [17,18]. Older adults were considered vulnerable to poor outcomes from SARS-CoV-2 infection. Results from 1970 COVID-19 patients in a nursing home in King County, Washington, suggested that residents with asymptomatic SARS-CoV-2 infection, especially those who had underlying comorbidities such as chronic cardiovascular and respiratory diseases in the nursing home setting, were at an increased risk of death during follow-up [19].

Asymptomatic SARS-CoV-2 infection has increased the difficulty and challenges of community COVID-19 prevention and control. In addition, asymptomatic patients usually have fewer opportunities and probability to seek medical treatment, resulting in some potential long-term health effects that are easily ignored. Therefore, it is important to pay more attention to the long-COVID-19 symptoms of asymptomatic SARS-CoV-2 infection patients. Although the long-term effects of COVID-19 have become an area of global concern currently, there are very limited published studies exploring the long-term consequences of asymptomatic SARS-CoV-2 infections, or the comparison of long-COVID-19 between symptomatic and asymptomatic initial SARS-CoV-2 infections. Thus, in this systematic review and meta-analysis, we aimed to review the data available to explore the long-term consequences of COVID-19 among those asymptomatic SARS-CoV-2 infection patients in the real world.

## 2. Methods

### 2.1. Search Strategy and Selection Criteria

This systematic review, and meta-analysis, was registered on PROSPERO (CRD42022367200). The study was strictly performed according to the Preferred Reporting Items for Systematic Reviews and Meta-Analyses (PRISMA, in the Appendix A) [20]. We conducted a systematic search in six databases, including PubMed, Embase, Web of Science, Science Direct, bioRxiv, and medRxiv from database inception to 14 October 2022 without language restrictions. The search terms were: ((post COVID-19) OR (long COVID-19) OR ((COVID-19 OR SARS-CoV-2 OR coronavirus) AND ((long-term effect) OR sequelae OR (post condition) OR (post syndrome) OR (long-term consequence)))) AND (asymptomatic OR symptomless OR nonsymptomatic). We used EndNoteX8.2 (Thomson Research Soft, Stanford, CA, USA) to manage records, screen, and exclude duplicates.

We included studies that examined the long-term consequences of asymptomatic SARS-CoV-2 infections. Asymptomatic individuals with positive test results for SARS-CoV-2 (asymptomatic infections) were defined as those who did not present any symptoms at the time of SARS-CoV-2 testing or diagnosis [4]. The following studies were excluded: (1) studies irrelevant to the subject of the meta-analysis, for example, those did not use asymptomatic SARS-Cov-2 infection as the exposure; (2) incomplete or inexact quantitative data provided for COVID-19 long-term consequences; (3) duplicate studies or overlapping participants; (4) reviews, editorials, conference papers, case report or series study, and animal experiments; (5) insufficient follow-up time (less than 3 months); and (6) the identification of COVID-19, such as the confirmed diagnosis of COVID-19 via reverse-transcription polymerase chain reaction (rt-PCR) test, serologic test or other means not mentioned in the text.

Studies were identified by two investigators (MYR and DJ) independently following the criteria above, while discrepancies were solved by consensus or with a third investigator (LQ).

### 2.2. Data Extraction

The extracted data included: (1) study characteristics, including the first author, article type, study design, publication time, the location where the study was conducted, and time range for inclusion of participants; (2) participant characteristics, including sample size, median age, sex ratio, body mass index (BMI), follow-up time, comorbidity, smoking status, the severity of COVID-19, and the dominant variant of concern (VOC) of SARS-Cov-2 when participants included. If the dominant VOC was not indicated in the text of the included study, investigators would search for it through Global Initiative on Sharing All Influenza Data (GISAID) system [21]. We recorded the time range of SARS-CoV-2 infection confirmed in the included study subjects, and used the GISAID system to search the epidemic SARS-CoV-2 strains in the country or region where the study was conducted during this period. The period was defined as a non-VOCs period if no VOC was prevalent, otherwise it was defined as VOCs period; and (3) the type and prevalence of COVID-19 long-term consequences. Data extraction and determination of information eligibility were conducted by two investigators (MYR and DJ) independently following the criteria above, while discrepancies were solved by consensus or with a third investigator (LQ).

### 2.3. Quality Assessment and Risk of Bias

The quality assessment and risk of bias were assessed using the Newcastle–Ottawa quality assessment scale for the included cohort studies. Cohort studies were classified as having low (≥7 stars), moderate (5–6 stars), and high risk of bias (≤4 stars) with an overall quality score of 9 stars. Quality assessment was conducted by two investigators (MYR and DJ) independently, while discrepancies were solved by consensus or with a third investigator (LQ). The publication bias was assessed using Egger’s test, and a *p* < 0.05 was considered as having publication bias.

### 2.4. Data Synthesis and Statistical Analysis

All analyses in this meta-analysis were conducted using Stata version 16.0 (Stata Corp, College Station, Texas, USA). Pooled prevalence (PP) with a corresponding 95% confidence interval (CI) was reported for the prevalence of long-term consequences of SARS-CoV-2 infections. Among asymptomatic SARS-CoV-2 infection cases, subgroup analysis was carried out for the core consequence of at least one symptom by age (children were defined as under 18 years old and adults as more than 18 years old), sample size, the country where the study was conducted, follow-up time and dominant VOC period. Odds ratio (OR) with a corresponding 95% CI was reported to compare the difference in the long COVID-19 consequences between asymptomatic infections and symptomatic infections.

On the basis of heterogeneity between estimates (I²), we used random-effects or fixed-effects models to pool the rates and adjusted estimates across studies separately. If I² ≤ 50%, we would use a fixed-effects model, indicating low to moderate heterogeneity. If I² ≥ 50%, which means significant heterogeneity, we would use random-effects models and estimate the tau square by the Dersimonian–Laird method.

## 3. Results

### 3.1. Basic Characteristics

We retrieved a total of 2707 potential articles in the databases up to 14 October 2022 in the original literature search (1859 in Web of Science, 307 in Embase, 333 in ScienceDirect, 208 in PubMed, 0 in BioRxiv, and 0 in MedRxiv). After removing 696 duplicate articles, we reviewed the titles and abstracts of the remaining 2011 articles, 1921 articles were excluded according to the inclusion and exclusion criteria. Of the 90 studies after full-text reading, 85 studies were excluded (76 were irrelevant to the subject of the meta-analysis and nine had insufficient data or data that could not be extracted). Finally, we included five published studies in this meta-analysis based on the inclusion criteria [14,22,23,24,25]. The flow chart study selection is shown in Figure 1.

Of the five studies included, most were limited to Europe (two in Italy, one in Germany, and one in Luxembourg), followed by South America (one in Brazil). Among the five included studies, the participants of three studies were adults and two were children, and most study (n = 4) cohorts were followed up for more than six months. The included studies described the long-term consequences of asymptomatic SARS-CoV-2 infection or compared long-term consequences of different severity of SARS-CoV-2 infection, involving a total of 1643 cases, including 597 cases of asymptomatic and 1046 cases of symptomatic SARS-CoV-2 infection. The characteristics of the included studies are shown in Appendix A.

### 3.2. Pooled Prevalence (PP) of Long-Term Consequences of Asymptomatic SARS-CoV-2 Infection

We calculated the PPs of 17 long-term consequences of asymptomatic SARS-CoV-2 infection. The asymptomatic SARS-CoV-2 infection could develop a variety of sequelae in different physical systems. The PP was 17.13% (95%CI, 7.55–26.71%) for at least one symptom, 15.09% (95%CI, 5.46–24.73%) for loss of taste, 14.14% (95%CI, −1.32–29.61%) for loss of smell, and 9.33% (95%CI, 3.07–15.60) for fatigue. Other long-term health effects such as dyspnoea, headache, psychiatric disorders et al. could also be observed in our analysis results. No one with an asymptomatic infection reported chest pain in the included studies. More analysis results of other sequelae are shown in Table 1.

We also compared the PPs for long-term consequences of asymptomatic SARS-CoV-2 infection between adults and children. Among adults with asymptomatic SARS-CoV-2 infection, 21.38% developed at least one sequela, the PP was 15.20% for fatigue, 6.82% for muscle or joint pain, 6.82% for dyspnea, 6.82% for cough, and 4.55% for headache. Among children with asymptomatic SARS-CoV-2 infection, 15.20% developed at least one sequela, the PP was 5.19% for fatigue, 4.44% for dyspnea, and 4.44% for headache. Some sequelae were observed in children but not among adults, such as physical disorders, neurological symptoms, loss of smell/test, and gastrointestinal. More analysis results are shown in Figure 2.

### 3.3. Subgroup Analysis of Pooled Prevalence (PP) of at Least One Symptom among Asymptomatic SARS-CoV-2 Infections by Sample Size, Country, Follow-Up Time, and Epidemic Period

In addition to age, we also conducted a subgroup analysis of PP for at least one symptom among asymptomatic SARS-CoV-2 infection cases by sample size, country, follow-up time, and epidemic period. The PP for at least one long-term symptom was 21.27% in the group with a sample size < 100 and 11.53% in the group with a sample size ≥ 100. As for the subgroup by country, the PP for at least one long-term symptom was 22.22% in Luxembourg, 38.64% in Germany, and 9.05% in Italy. The PP for at least one long-term symptom was 11.35% in the less-than-12-months follow-up group and 38.64% in the more-than-12-months follow-up group. The PP of at least one long-term symptom after asymptomatic SARS-CoV-2 infection was 15.20% in the non-VOC period, and 21.38% in the VOC period. More details are shown in Table 2.

### 3.4. Comparison of Long-Term Consequences of Asymptomatic and Symptomatic SARS-CoV-2 Infection

Among total symptomatic SARS-CoV-2 infection cases, 53.02% developed at least one symptom, 48.19% of the mild cases, and 66.54% of the moderate/severe cases developed at least one symptom. The PPs were 22.07% for fatigue, 21.00% for muscle or joint pain, 13.78 for dyspnea, 11.97% for neurological symptoms, and 10.61% for loss of taste or smell among total symptomatic SARS-CoV-2 infection cases. The PPs of the remaining consequences were less than 10%. Our analysis results suggested that the long-term consequences of SARS-CoV-2 infection were more frequent in symptomatic cases than those in asymptomatic cases, whether they were mild or moderate/severe cases. More analysis results about the PPs for the long-term consequences of different severity of COVID-19 are shown in Table 3.

Compared with symptomatic SARS-CoV-2 infection, asymptomatic infection was associated with a lower risk of developing COVID-19 sequelae, such as developing at least one symptom (OR = 0.20, 95%CI, 0.09–0.45), fatigue (OR = 0.19, 95%CI, 0.08–0.49), cutaneous lesions (OR = 0.26, 95%CI, 0.09–0.72), neurological symptoms (OR = 0.15, 95%CI, 0.07–0.30), dyspnoea (OR = 0.25, 95%CI, 0.07–0.30), muscle or joint pains (OR = 0.20, 95%CI, 0.10–0.38), loss of taste/smell (OR = 0.10, 95%CI, 0.02–0.58) et al., as shown in Table 4 and Figure 3. We also conducted a subgroup analysis for some long-term consequences of asymptomatic and symptomatic SARS-CoV-2 infection based on the age group, seen in Appendix A.

### 3.5. Quality Evaluation and Publication Bias

We evaluated the quality of the included cohort studies according to the Newcastle–Ottawa quality assessment scale, all of them were of good quality and had a low risk of bias (≥7 stars), as shown in Appendix A. We assessed the publication bias of the PP of at least one symptom in asymptomatic cases and symptomatic cases using Egger’s test, the results suggested there was no publication bias (*p* > 0.05).

## 4. Discussion

Since the outbreak of COVID-19 in 2019, it has brought burdens and challenges to more than 200 countries and regions around the world. A growing number of studies have found that a significant number of people infected with SARS-CoV-2 were asymptomatic [4]. Therefore, evaluating the long-term consequences of asymptomatic SARS-CoV-2 infections will not only help to formulate more reasonable plans for their treatment and care, but also help to provide a more scientific basis for the control and management of the COVID-19 epidemic.

Our systematic review and meta-analysis of five articles, involving a total of 1643 cases, including 597 cases of asymptomatic and 1043 cases of symptomatic SARS-CoV-2 infection, estimated the PPs for 17 kinds of long-term COVID-19 consequences. A subgroup analysis was carried out for the core consequence of at least one symptom by age, sample size, the country where the study was conducted, follow-up time, and the dominant VOC period. We also estimated the pooled OR long-term consequences between asymptomatic and symptomatic SARS-CoV-2 infections. This was the first systematic review and meta-analysis describing the long-term consequences of asymptomatic SARS-CoV-2 infections.

Available data showed that in the cases of asymptomatic SARS-CoV-2 infection, 17.13% had one or more COVID-19-related long-term symptoms, which were mild and acceptable. Among asymptomatic SARS-CoV-2 infection cases, adults had a higher proportion of long-COVID-19 than children (21.38% and 15.20%, respectively), such as cough, dyspnea, and so on, which is similar results seen in previous studies [22]. There was evidence that there were no significant differences between the SARS-CoV-2 RNA loads between adults and children [26]. Adults with asymptomatic SARS-CoV-2 infection faced a higher PP of COVID-19-related long-term consequences, which may be related to increased viral entry mediators such as ACE2 and TMPRSS2 in respiratory epithelial cells, and increased pro-inflammatory cytokine production [27,28]. In addition, in the lung injury mice model induced by lipopolysaccharide (LPS), the lung barrier function of neonatal mice was better preserved in the face of inflammatory lung injury, suggesting that the lung barrier function of adults and children may be different [29].

The age distribution of long-term COVID-19 consequences in children was different. An Italian cohort study found that the cumulative incidence of COVID-19 increased from 18.3% in children aged 0–5 years to 34.4% in children aged 11–16 years [22]. This trend was confirmed in most long COVID-19-related disorders, while respiratory diseases were inversely associated with age, and the incidence of gastrointestinal and dermatological disorders was not significantly affected by age [22]. Olive Tang et al. found that all nursing home residents (average age at least 70 years) infected with SARS-CoV-2 had a very high risk of hospitalization and death, even if asymptomatic [19]. At present, data on asymptomatic COVID-19 infection in children and the elderly are still limited. In the future, more original studies on asymptomatic SARS-CoV-2 infections in old people and children of different ages are needed to provide more scientific prevention or treatment recommendations to reduce the burden of COVID-19 on them.

The available data showed that asymptomatic infected cases had a significantly reduced risk of long-term COVID-19 consequences compared to symptomatic infected cases. There is an increasing amount of evidence showing that the long-term consequences of COVID-19 are related to the severity of SARS-CoV-2 infection at the time of onset, that is, the more severe the disease, the higher the risk, and the greater the severity, of long-COVID [22,30]. Maddalena Peghin et.al found that a proportional increase in the number of symptoms during the onset of COVID-19 was one of the independent risk factors for post-COVID-19 syndrome [14]. At present, an understanding of COVID-19 is still developing, and the mechanism of asymptomatic infection of COVID-19 is still not clear, which needs to be improved in the future.

The best way to fight against the long-term consequences of COVID-19 is to avoid getting infected with SARS-CoV-2, and vaccination remains one of the most significant tools. The coverage of vaccination against COVID-19 in asymptomatic infections remains low, with only 6.02% (108/1795) in a Scottish study [9]. Available data showed that the longer the follow-up period, the higher the proportion of asymptomatic infections with at least one long-term COVID-19 symptom. On one hand, some symptoms may be more insidious in the early stage, on the other hand, the symptoms may be related to the reduction of protective antibody titers in the body [31]. Vaccination can not only effectively increase neutralizing antibody titers [32], but also reduce the risk of seven sequelae, including changes in smell, change in taste, hearing problems, poor appetite, balance problems, confusion or difficulty concentrating, and anxiety or depression [9]. The current evidence on the effects of vaccination on post-COVID-19 conditions in asymptomatic infections is still limited, and more related original studies are needed in the future to provide a better basis for expanding vaccination coverage.

Our study has several limitations. First, it may be more difficult to collect information about symptoms from children than adults, which does not rule out the existence of bias. Second, due to the limited data, the age range of the children’s group was relatively broad, and we added more detailed information about long-COVID for children of different ages in the discussion section. Third, since the absence of some consequences might not be recorded in some works, there was probably a bias in the analysis. Fourth, we should consider the absence of data on asymptomatic cases, since many of those may not be recognized as COVID-19 in the original studies.

## 5. Conclusions

In conclusion, patients might experience some long-term health effects with different symptoms after asymptomatic SARS-CoV-2 infection, and the prevalence of different symptoms might vary in children and adults, but the overall level of occurrence was not high. Our results suggested that the long-term effects of asymptomatic SARS-CoV-2 infection exist, but the risk and frequency of developing long-term symptoms in asymptomatic SARS-CoV-2 infected persons are significantly lower than symptomatic SARS-CoV-2 infection cases. These results should play an important role in guiding the long-term health care of post-SARS-CoV-2 infected patients and the rational allocation of medical resources.

## Figures and Tables

**Figure 1 ijerph-20-01613-f001:**
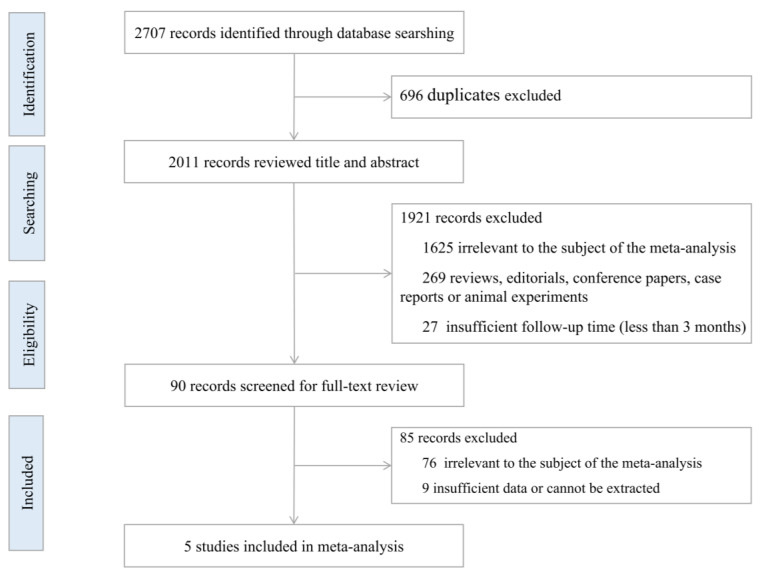
Flowchart of the study selection.

**Figure 2 ijerph-20-01613-f002:**
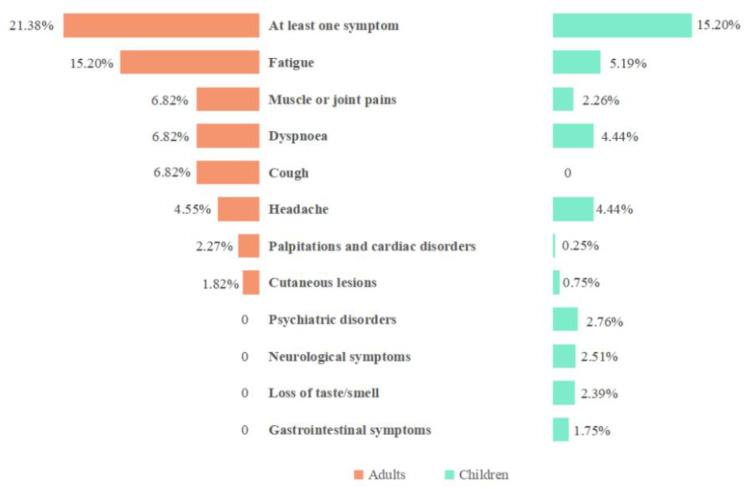
Comparison of pooled prevalence (PP) of long-term consequences of asymptomatic SARS-CoV-2 infection among adults and children.

**Figure 3 ijerph-20-01613-f003:**
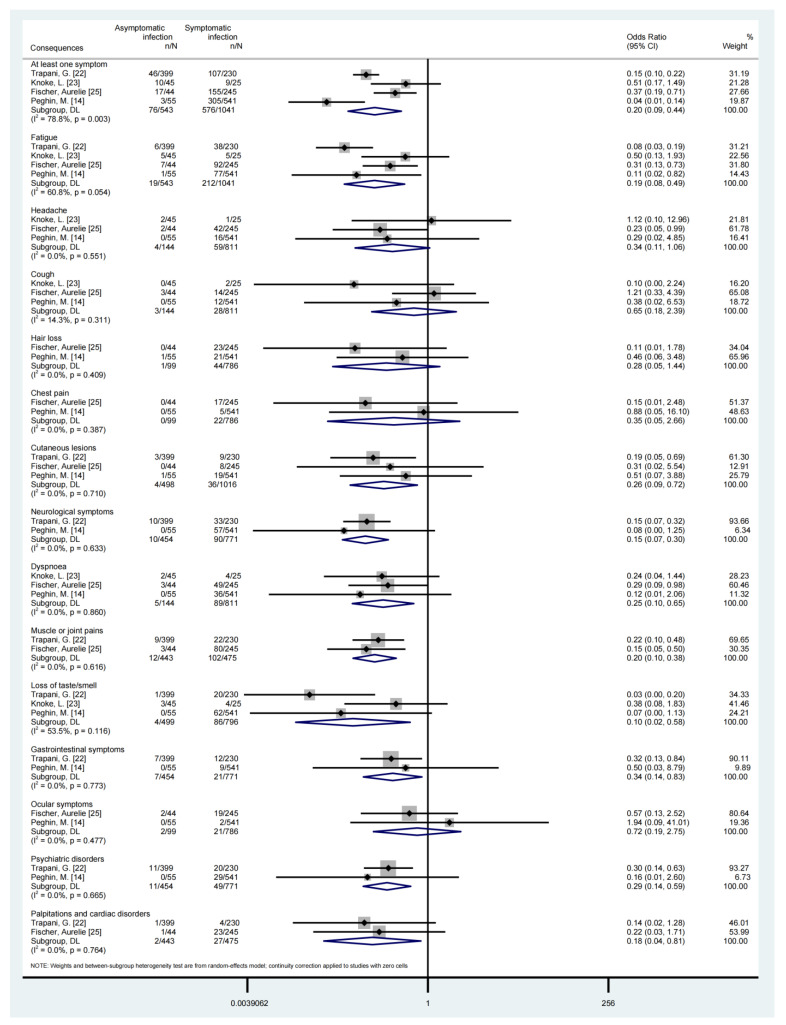
Forest plot of long-term consequences of asymptomatic or symptomatic SARS-CoV-2 infection [14,22,23,25].

**Table 1 ijerph-20-01613-t001:** Pooled prevalence (PP) of long-term consequences of asymptomatic SARS-CoV-2 infection.

Consequences	Data Source	Patients n/N	PP(%)	95%CI(%)	*p*-Value	I^2^(%)	*p*-Heterogeneity
At least one symptom	1, 2, 4, 5	76/543	17.13	7.55–26.71	<0.05	85.40	<0.05
Loss of taste	3, 4	8/97	15.09	5.46–24.73	<0.05	-	-
Loss of smell	3, 4	15/97	14.14	−1.32–29.61	>0.05	81.00	<0.05
Fatigue	1, 2, 3, 4, 5	35/596	9.33	3.07–15.60	<0.05	87.00	<0.05
Cough	2, 4, 5	3/144	6.82	−0.63–14.27	>0.05	-	-
Dyspnoea	4, 5	5/144	5.38	0.70–10.07	<0.05	0.00	>0.05
Ocular symptoms	4, 5	2/99	4.55	−1.61–10.70	>0.05	-	-
Headache	2, 4, 5	4/144	4.49	0.19–8.80	<0.05	0.00	>0.05
Muscle or joint pains	1, 4	12/443	3.11	−0.32–6.35	>0.05	28.00	>0.05
Psychiatric disorders	1, 5	11/454	2.76	1.15–4.36	<0.05	-	-
Neurological symptoms	4, 5	10/454	2.51	0.97–4.04	<0.05	-	-
Loss of taste/smell	1, 2, 5	4/499	2.39	−3.54–8.31	>0.05	66.30	>0.05
Hair loss	4, 5	1/99	1.82	−1.71–5.35	>0.05	-	-
Gastrointestinal symptoms	1, 5	7/454	1.75	0.47–3.04	<0.05	-	-
Cutaneous lesions	4, 5	4/498	0.81	−0.01–1.63	>0.05	0.00	>0.05
Palpitations and cardiac disorders	1, 4	2/443	0.28	−0.21–0.76	>0.05	0.00	>0.05
Chest pain	4, 5	0/99	-	-	-	-	-

**Table 2 ijerph-20-01613-t002:** Subgroup analysis of pooled prevalence (PP) of at least one symptom among asymptomatic SARS-CoV-2 infection cases by age group, sample size, country, follow-up time, and epidemic period.

Subgroups		Data Source	Patients n/N	PP(%)	95%CI(%)	*p*-Value	I^2^(%)	*p*-Heterogeneity
Age group	Adults	1, 2, 4, 5	20/99	21.38	−11.12–53.87	>0.05	94.30	<0.05
	Children		56/444	15.20	5.25–25.15	<0.05	64.20	>0.05
Sample size	<100	1, 2, 4, 5	30/144	21.27	1.71–40.83	<0.05	90.20	<0.05
	≥100		46/399	11.53	8.40–14.66	<0.05	-	-
Country	Italy	1, 2, 4, 5	49/454	9.05	3.20–14.90	<0.05	67.70	>0.05
	Germany		10/45	22.22	10.08–34.37	<0.05	-	-
	Luxembourg		17/44	38.64	24.25–53.02	<0.05	-	-
Follow-up time	<12 months	1, 2, 4, 5	59/499	11.35	4.76–17.95	<0.05	69.90	<0.05
	≥12 months		17/44	38.64	24.25–53.02	<0.05	-	-
Epidemic period	non-VOCs period	1, 2, 4, 5	56/444	15.20	5.25–25.15	<0.05	64.20	>0.05
	VOCs period		20/99	21.38	−11.12–53.87	>0.05	94.30	<0.05
Overall		1, 2, 4, 5	76/543	17.13	7.55–26.71	<0.05	85.40	<0.05

**Table 3 ijerph-20-01613-t003:** Pooled prevalence (PP) of long-term consequences in patients with different severity of COVID-19.

Consequences	Severity of COVID-19 Disease			
	Asymptomatic Infection	Symptomatic Infection		
		Total	Mild Cases	Moderate/Severe Cases
At least one symptom	17.13 (7.55–26.71) *	53.02 (44.54–61.49) *	48.19 (32.78–63.60) *	66.54 (34.86–98.23) *
Fatigue	9.33 (3.07–15.60) *	22.07 (11.18–32.95) *	20.48 (1.42–39.54) *	40.05 (4.39–75.71) *
Headache	4.49 (0.19–8.80) *	8.06 (−1.78–17.90)	8.65 (−5.44–22.73)	12.20 (−6.36–30.76)
Loss of smell	14.14 (−1.32–29.61)	-	-	-
Cough	6.82 (−0.63–14.27)	4.02 (0.88–7.15) *	2.15 (0.67–3.62) *	7.75 (−2.28–17.78)
Hair loss	1.82 (−1.71–5.35)	6.38 (1.01–11.76) *	6.49 (0.79–12.19) *	4.13 (1.19–7.06) *
Loss of taste	15.09 (5.46–24.73) *	-	-	-
Chest pain	-	3.73 (−2.15 -9.61)	2.36 (−1.38–6.10)	7.85 (−5.60–21.30)
Cutaneous lesions	0.81 (−0.01–1.63)	3.53 (2.40–4.67) *	3.60 (2.09–5.11) *	2.80 (0.36–5.25) *
Neurological symptoms	2.51 (0.97–4.04) *	11.97 (8.35–15.59) *	-	-
Dyspnoea	5.38 (0.70–10.07) *	13.78 (2.97–24.60) *	9.84 (0.49–19.20) *	23.97 (−5.66–53.59)
Muscle or joint pains	3.11 (−0.32–6.35)	21.00 (−1.63–43.62)	-	-
Loss of taste/smell	2.39 (−3.54–8.31)	10.61 (8.47–12.75) *	-	-
Gastrointestinal symptoms	1.75 (0.47–3.04) *	3.18 (−0.27–6.63)	-	-
Ocular symptoms	4.51 (−1.61–10.70)	3.87 (−3.36–11.10)	3.12 (−2.57–8.81)	12.07 (3.69–20.45) *
Psychiatric disorders	2.76 (1.15–4.36) *	6.65 (3.47–9.84) *	-	-
Palpitations and cardiac disorders	0.28 (−0.21–0.76)	5.39 (−2.10–12.87)	-	-

* *p*-value < 0.05.

**Table 4 ijerph-20-01613-t004:** Comparison of long-term consequences of asymptomatic and symptomatic SARS-CoV-2 infection.

Consequences	Data Source	Asymptomatic Infection n/N	Symptomatic Infection n/N	OR	95%CI	*p*-Value	I^2^ (%)	*p*-Heterogeneity
At least one symptom	1, 2, 4, 5	76/543	576/1041	0.20	0.09–0.45	<0.05	79.40	<0.05
Fatigue	1, 2, 4, 5	19/543	212/1041	0.19	0.08–0.49	<0.05	61.30	<0.05
Headache	2, 4, 5	4/144	59/811	0.34	0.12–1.06	>0.05	0.00	>0.05
Cough	2, 4, 5	3/144	28/811	0.64	0.17–2.43	>0.05	17.10	>0.05
Hair loss	4, 5	1/99	44/786	0.28	0.05–1.45	>0.05	0.00	>0.05
Chest pain	4, 5	0/99	22/786	0.35	0.05–2.66	>0.05	0.00	>0.05
Cutaneous lesions	1, 4, 5	4/498	36/1016	0.26	0.09–0.72	<0.05	0.00	>0.05
Neurological symptoms	1, 5	10/454	90/771	0.15	0.07–0.30	<0.05	0.00	>0.05
Dyspnoea	2, 4, 5	5/144	89/811	0.25	0.10–0.65	<0.05	0.00	>0.05
Muscle or joint pains	1, 4	12/443	102/475	0.20	0.10–0.38	<0.05	0.00	>0.05
Loss of taste/smell	1, 2, 5	4/499	86/796	0.10	0.02–0.58	<0.05	53.50	>0.05
Gastrointestinal symptoms	1, 5	7/454	21/771	0.34	0.14–0.83	<0.05	0.00	>0.05
Ocular symptoms	4, 5	2/99	21/786	0.72	0.19–2.75	>0.05	0.00	>0.05
Psychiatric disorders	1, 5	11/454	49/771	0.28	0.14–0.59	<0.05	0.00	>0.05
Palpitations and cardiac disorders	1, 4	2/443	27/475	0.18	0.04–0.81	<0.05	0.00	>0.05

## Data Availability

Data are available from the corresponding author by request.

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
