# Peer review of "Long-Term Consequences of Asymptomatic SARS-CoV-2 Infection: A Systematic Review and Meta-Analysis"

_ijerph, 2023, doi:10.3390/ijerph20021613_

Round 1

Reviewer 1 Report

Yurui Ma et al. proposed a systematic review and meta-analysis on long-term consequences of asymptomatic SARS-Cov-2 infection. The idea is interesting, but several issues should be better clarified.

1. "The data to be were extracted included"   ---> please review this sentence

2. "The following studies were excluded: The following studies were excluded:"  ---> please, remove duplicated sentence

3. If the dominant VOC was not indicated in the 140 text of the included study, investigators would search for it through Global Initiative on 141 Sharing All Influenza Data (GISAID) system  ----> please, explain better the use of GISAID

4. "Children were defined as under 18 years old 159 and adults as more than 18 years old"  ---> Would it be useful to classify minors into children (<12) and adolescents (12-18)?

5. In the search strategy you have indicated 13 October... then, in the Results 14 October...

6. BioRxiv and MedRxiv are only mentioned in the search strategy but not in the results

7. The box with the 696 duplicates excluded is missing in the flow chart

8. It would be interesting to be able to include the 13 non-full-text papers in your work. That is a lot of work and would greatly improve the results 

9. You write: "and most study cohorts were followed up..." how many? 

10. 1643 cases .... 597 asymptomatic and 1043 symptomatic. what's about the other 3?

11. Loss of smell is 14.14% (95%CI is missing)

12. You reported chest pain 0/99 considering two data sources, only because obviously this consequence was mentioned (and was absent) in these two papers... But there is a bias in the analysis, since the absence of some consequences may be not recorded in some works... Chest pain may also be absent in the other papers, although it was not mentioned... Similarly, I would like to ask you if there are other consequences that have been recorded as 'absent' in some works.

Please, consider whether is it possible include as outcomes only those consequences mentioned by all the papers.

13. Among the limitations, consider the absence of data on asymptomatic cases, precisely because many of these were not even recognised as Covid-19.

Author Response

1. "The data to be were extracted included"   ---> please review this sentence

Response: Thanks for the reviewer’s suggestion. We have revised the sentence.

2. "The following studies were excluded: The following studies were excluded:"  ---> please, remove duplicated sentence

Response: Thanks for the reviewer’s suggestion. We have removed duplicated sentence.

3. If the dominant VOC was not indicated in the text of the included study, investigators would search for it through Global Initiative on Sharing All Influenza Data (GISAID) system  ----> please, explain better the use of GISAID

Response: Thanks for the reviewer’s suggestion. We recorded the time range of SARS-CoV-2 infection confirmed of the included study subjects, and used the GISAID system to search the epidemic SARS-CoV-2 strains in the country or region where the study conducted during this period. The period was defined as a non-VOCs period if no VOC was prevalent, otherwise it was defined as VOCs period. We have added more details in the Methods section.

4. "Children were defined as under 18 years old 159 and adults as more than 18 years old"  ---> Would it be useful to classify minors into children (<12) and adolescents (12-18)?

Response: Thanks for the reviewer’s suggestion. Due to the limited data, the age range of the children’s group was relatively broad, and it is difficult to classify minors. We added more detailed information about long-COVID for children of different ages in the discussion section.

5. In the search strategy you have indicated 13 October... then, in the Results 14 October...

Response: Thanks for the reviewer’s suggestion. It was 14 October. We have revised it in the search strategy section.

6. BioRxiv and MedRxiv are only mentioned in the search strategy but not in the results

Response: Thanks for the reviewer’s suggestion. It was 14 October. 0 potential articles was searched in BioRxiv and MedRxiv databases. We have added it in the Results section.

7. The box with the 696 duplicates excluded is missing in the flow chart

Response: Thanks for the reviewer’s suggestion.We have revised the flow chart. 

8. It would be interesting to be able to include the 13 non-full-text papers in your work. That is a lot of work and would greatly improve the results.

Response: Thanks for the reviewer’s suggestion. We obtained the text of the 13 studies by contacting the authors, and found 9 of them were irrelevant to the subject of the meta-analysis, and the data of other 4 studies were insufficient or cannot be extracted. We have updated the flow chart of the study selection (Figure 1) and revised the manuscript.

9. You write: "and most study cohorts were followed up..." how many?

Response: Thanks for the reviewer’s suggestion. 4 of 5 included study were followed up for more than 6 months. We have added the details in the manuscript.

10. 1643 cases .... 597 asymptomatic and 1043 symptomatic. what's about the other 3?

Response: Thanks for the reviewer’s suggestion. We have revised, there were actually 1643 cases, including 597 cases of asymptomatic and 1046 cases of symptomatic SARS-CoV-2 infection.

11. Loss of smell is 14.14% (95%CI is missing)

Response: Thanks for the reviewer’s suggestion. We have added the 95%CI of the PP for smell in the manuscript.

12. You reported chest pain 0/99 considering two data sources, only because obviously this consequence was mentioned (and was absent) in these two papers... But there is a bias in the analysis, since the absence of some consequences may be not recorded in some works... Chest pain may also be absent in the other papers, although it was not mentioned... Similarly, I would like to ask you if there are other consequences that have been recorded as 'absent' in some works.Please, consider whether is it possible include as outcomes only those consequences mentioned by all the papers. 

Response: Thanks for the reviewer’s suggestion. We extracted data of consequences from the text, tables, figures, and supplement materials of the included studies. Due to differences in study methods and content, different studies evaluated and focused on different indicators and consequences, so some consequences were recorded in some studies, but might not be mentioned in other articles, and we could not know the specific situation of these unmentioned consequences (absent or not). In addition to chest pain, we did notice several other outcomes that were recorded as ‘absent’ in some works but not mentioned in others. Because of the limited available literature, we pooled as many studies as possible to more fully assess the long-term consequences of asymptomatic SARS-CoV-2 infection. Since the absence of some consequences may be not recorded in some works, there was a bias in the analysis. It is really a good idea to consider only those consequences mentioned by all included papers, but unfortunately it is a little difficult to achieve. Therefore, we have added this as a limitation in the discussion section.

13. Among the limitations, consider the absence of data on asymptomatic cases, precisely because many of these were not even recognised as Covid-19.

Response: Thanks for the reviewer’s suggestion. We have added it as a limitation in the Discussion section.

Author Response

1. In the abstract section (line 25), authors mentioned “Our results suggested that there were long-term effects of asymptomatic SARS-CoV-2 infection”__ I emphatically suggest that authors need to give example here (Although, the authors may consider the number of words allowed by the journal).

Response: Thanks for the reviewer’s suggestion. We have added some examples in the abstract section.

2. Line 122-123- just be reminded to check the duplication mentioned about exclusion criteria. “Studies excluded”…

Response: Thanks for the reviewer’s suggestion. We have removed the duplicated sentence.

3. Though, it was discussed in the study limitation that age range of the children’s group was relatively broad, however, as a reader, I appreciate more if you could give the age range that was considered in the study in the criteria.

Response: Thanks for the reviewer’s suggestion. We considered children as under 18 years old, and adults as more than 18 years old. We have added the age range that was considered in the study in the data synthesis and statistical analysis section.

4. Please include the time range of the studiesreviewed in the manuscript. 

Response: Thanks for the reviewer’s suggestion. We have included the time range of the studies reviewed in the manuscript in the Supplementary table 1.

5. Suggest to discuss the presence or absence of publication bias in this study.

Response: Thanks for the reviewer’s suggestion. We have added the results of publication bias in the Results section.